# Electroactive 3D Printed Scaffolds Based on Percolated Composites of Polycaprolactone with Thermally Reduced Graphene Oxide for Antibacterial and Tissue Engineering Applications

**DOI:** 10.3390/nano10030428

**Published:** 2020-02-28

**Authors:** Carolina Angulo-Pineda, Kasama Srirussamee, Patricia Palma, Victor M. Fuenzalida, Sarah H. Cartmell, Humberto Palza

**Affiliations:** 1Department of Chemical Engineering and Biotechnology and Materials, University of Chile, Santiago 8370456, Chile; 2Millenium Nuclei in Soft Smart Mechanical Metamaterials, Universidad de Chile, Santiago 8370456, Chile; 3Department of Biomedical Engineering, Faculty of Engineering, King Mongkut’s Institute of Technology Ladkrabang (KMITL), Bangkok 10520, Thailand; kasama.sr@kmitl.ac.th; 4Department of Pathology and Oral Medicine, University of Chile, Santiago 8380492, Chile; ppalma@odontologia.uchile.cl; 5Department of Physics, University of Chile, Santiago 8370449, Chile; vfuenzal@ing.uchile.cl; 6Department of Materials, The University of Manchester, Manchester M13 9PL, UK; sarah.cartmell@manchester.ac.uk

**Keywords:** electroactive biomaterials, conductive polymers, 3D scaffolds, electrical stimulation, bioelectric effects, antibacterial properties

## Abstract

Applying electrical stimulation (ES) could affect different cellular mechanisms, thereby producing a bactericidal effect and an increase in human cell viability. Despite its relevance, this bioelectric effect has been barely reported in percolated conductive biopolymers. In this context, electroactive polycaprolactone (PCL) scaffolds with conductive Thermally Reduced Graphene Oxide (TrGO) nanoparticles were obtained by a 3D printing method. Under direct current (DC) along the percolated scaffolds, a strong antibacterial effect was observed, which completely eradicated *S. aureus* on the surface of scaffolds. Notably, the same ES regime also produced a four-fold increase in the viability of human mesenchymal stem cells attached to the 3D conductive PCL/TrGO scaffold compared with the pure PCL scaffold. These results have widened the design of novel electroactive composite polymers that could both eliminate the bacteria adhered to the scaffold and increase human cell viability, which have great potential in tissue engineering applications.

## 1. Introduction

The electroactive biomaterials are smart systems, which are able to deliver electrical stimulation (ES) to the surrounding media to impart an effect on the behavior of biological systems [1,2]. In particular, these biomaterials take advantage of the effect of a direct current (DC) or an electrical field on both cell proliferation and differentiation, stimulating, for instance, the regeneration of muscles, organs, and/or bones [3,4,5,6,7,8]. Moreover, ES has been studied as a potential tool in tissue regeneration to increase the proliferation and differentiation of human stem cells [5,9,10,11,12,13]. There are several reports in the literature on the use of ES for fracture treatments and tissue repairs using electrical stimulation methodologies, particularly DC or electromagnetic fields [14]. It was established that by applying an electric field (EF), cellular behavior can be modified, including the orientation, proliferation, and rate and direction of cell migration of the corneal, epithelial, and vascular cells, among others [15]. In this context, smart electroactive polymers, which are considered a new generation of intelligent materials, have been developed to transfer electrons/ions or to produce changes in charge distribution under a controlled EF, allowing the development of various areas in tissue engineering [6]. The advantage of applying an external electrical stimulus lies in its precise control of the magnitude, time, and periodicity of the voltage used, obtaining different effects according to the form of ES applied [16].

The reduction in bacterial growth under ES as selective biocidal effect has not been widely studied [1]. In particular, this electrical antibacterial effect has barely been studied in polymeric biomaterials, which presents an opportunity to complement several strategies developed to eradicate bacterial growth motivated by the increase in antibiotic resistance and the high cost of treating bacterial infections [17]. For example, osteomyelitis and other bacterial diseases associated with bone prostheses can bring serious consequences to patients, such as the removal of bone implants, lengthening the wound healing processes, and increasing the cost associated with surgery [17,18,19,20,21,22]. Therefore, several materials and methods have been developed to eradicate bacteria colonies from biomaterial surfaces before the formation of a biofilm, since it is harder to eliminate the consortium of bacterial colonies afterward [18]. In this context, ES applied directly onto the material surface appears to be one of the most effective antibacterial treatments against different types of bacterial biofilms [22,23]. The mechanism of the effect of electric current on bacterial cells are not yet fully understood, although it is associated with changes in the medium through: generation of biocidal ions and reactive oxygen species (ROS), pH modification, and electrophoretic forces [24,25]. It has been shown that DC can be used to modulate bacterial detachment and movement on conductive surfaces [26,27,28,29,30,31]. Negatively charged bacteria were electrically manipulated by modulating the electrophoretic movements under DC, inducing an antiadhesive effect [26]. As a result, the bacteria were pushed away from the electrically charged surface by electrophoretic effects and/or were released by electroosmotic flows which lead to the movement of the bacteria [26]. In addition, attached bacteria may be detached due to electron transfer between the bacterial membrane and the surface that allows the passage of current [32].

The most studied biomaterials for ES are those based on metal composites, whereas the polymers are less studied despite their great potential in tissue engineering applications [16,33]. The extrinsic conductive polymeric composites have the advantage that they can be made of a biocompatible polymer matrix reinforced with the biologically active and electrically conductive materials such as graphene-based particles [34]. These bioactive electrical polymer/graphene composites can be designed not only to provide electrical properties, but also a positive biological effect [35,36] due to the different types of functional groups that graphene derivatives present, which could interact with biological systems such as bacteria and human cells [37].

Based on this precedent, novel electroactive scaffolds were developed in this study based on the composites of Polycaprolactone (PCL) filled with conductive Thermally Reduced Graphene Oxide (TrGO) nanoparticles at above the percolation threshold. These electroactive scaffolds were 3D printed to control their porous structures. By applying a voltage along on the surface of the composite scaffolds, a bactericidal effect was observed. Moreover, under the same regime of ES, the adhesion and viability of human stem cells were further enhanced as compared with pure PCL scaffolds with and without ES.

## 2. Materials and Methods

### 2.1. Synthesis of Graphene Oxide (GO) and Thermally Reduced Graphene Oxide (TrGO)

High-purity graphite powder was used to obtain TrGO in two steps: Oxidation and thermal reduction. For the oxidation of the graphite, the Hummers and Offeman methodology was used [38]. In short, potassium permanganate and sodium nitrate were used to oxidize the graphite powder in concentrated sulfuric acid solution as described by the authors of Reference [39]. All reagents were obtained from Merck (Darmstadt, Germany). Later, the GO powder was vacuum dried at 60 °C and then milled to obtain fine GO powder. TrGO was then obtained by thermal shock at 1000 °C for 20 s in a vertical quartz reactor [40]. The advantage of thermal reduction is the ability to produce modified graphene sheets without the need for solvents [40].

### 2.2. Characterization of GO and TrGO Particles

The nanoparticles obtained from the methodology described above were analyzed by X-ray diffraction (XRD) with a scintillation detector with Bragg-Brentano geometry with a copper radiation source (Cu λ = 1.504 Å), (Bruker D8 Advance, Billerica, MA, USA), 2θ sweep from 2° to 40° was performed with a scanning rate of 0.02 °/s. Graphene-based particles were characterized by X-ray photoelectron spectroscopy (XPS, Physical Electronics 1257 model, Billerica, MA, USA). The samples were deposited onto the thin gold films through an aqueous dispersion (500 µL) of GO and TrGO particles. XPS was performed on these samples using Al nonmonochromatic radiation in the binding energy region of 300–280 eV belonging to C1s. The Au 4f photoelectron line was used as energy reference. The Multipack software was used for fitting the curves using a pseudo-Voigt adjustment (80% Gaussian and 20% Lorentzian) [41]. This allowed the identification C sp^2^ hybridized carbon and the bands attributed to the main functional groups [42]. Raman spectra were recorded on an alpha300R Witec confocal Raman microscope equipped with a 532-nm wavelength laser. The ratio of the D and G bands intensities was analyzed by Lorenztian fitting, and the density of defects in the graphite derivatives was obtained as described by the authors of Reference [43].

### 2.3. PCL/TrGO Composites

The composites were prepared by melt mixing at 50 revolutions per minute (rpm) and at 90 °C in the nitrogen environment using a Brabender Plasticorder (Brabender, Germany) for 40 min. After this process, the films of polymers materials were obtained using an electro-hydraulic press with control heating and fast cooling system electro (HP model D-50) at 90 °C for 2 min. The sample thickness was 2 mm for the scaffolds composites [44]. Then, 10% wt of TrGO was used to obtain the polymer composites. This concentration was selected based on printability and conductivity values of composites films with different loads of TrGO 5% wt, 10% wt, 15% wt, and 20% wt (Appendix A). Raman spectroscopy was used to analyze the composites and graphene-based particles. The films were prepared and cut to carry out the 3D-printing process at high temperatures (T_PCL_ = 170 °C and T_PCL/TrGO_ = 220 °C), whereas the thermogravimetric analysis (TGA instruments, Q50, TA) was performed between room temperature and 500 °C at 20 °C/min under nitrogen and air. The design of the 3D scaffolds was carried out using CAD based on the study of S. Mehendale et al. [45] in which different designs were proposed and mechanical properties and bone cell adhesion were evaluated. Based on these results, the angles between filaments that form the layers were 0°, 60°, and 120°. Using these framework angles, the pore size distribution agrees with the values required to work with human bone marrow-derived mesenchymal stem cells (hBM-SCs) between 250 and 400 µm [46]. During the printing process, a 3-M Scotch Blue Masking Tape was used to cover the platform and increase the adhesion of the first layer, which was necessary to provide stability to the five-layer structure. A 3D printer and high temperature accessories were used (Envionstec, 4th generation, Germany). The optimal printing parameters for PCL and its composites are summarized in Table 1. PCL has a high viscosity compared to other thermoplastics [47]. Therefore, it was necessary to print it at much higher temperature than the T_m_ = 60 °C (obtained by means of DSC graph, Appendix A) although the temperature needed to remain below the degradation temperature described below. In addition, nitrogen was used during the printing process, which allowed us to eject or push the molten material through the printing needle avoiding or delay the thermal degradation of the polymer.

The DC electrical resistance of the scaffold was measured using a source measure unit (Keysight B290) based on the ASTM D257 standard at three different sites (located on opposite corners). With these resistance values, the conductivity of the scaffolds was calculated considering the scaffold thickness (t) and effective area (A) of the measuring electrode (Keysight 11059A).

The angle of the contact was evaluated to quantify changes in hydrophobicity of the PCL/TrGO scaffolds with respect to the PCL scaffolds. This protocol was based on the standard test [48] and on recent articles for 3D scaffolds [48,49]. A drop shape analyzer (Krüss, DSA25, Hamburg, Germany) and Advance software (Krüss, Germany) were used to obtain the contact angle of the tested surfaces. A drop of 10 µL of milliQ deionized water was used in each test at three different times at 5 s, 30 s, and 60 s. (n = 5 independent tests with three measurements per test). An atomic force microscope (AFM) (Witec, alpha300, Ulm, Germany) was used to analyze two regions of the scaffolds and five different zones for each region. Root mean square (RMS) roughness values were obtained by Gwyddion software from these topographic images. From the rugosity distribution, the skewness (Rsk), and kurtosis (Rk) values were obtained to correlate the scaffolds topography with biological behavior without ES. Scanning electron microscopy (SEM) was used to analyze the structure of 3D-printed scaffolds using a LEO VP1400 from Zeiss. The PCL scaffolds were sputter coated with gold (10 nm) to produce a conductive surface before analysis.

### 2.4. Biological Assays under Direct Current

DC experiments were performed to demonstrate the antibacterial effect and biocompatibility with human cells of electroactive scaffolds under electrical stimulation. The experimental setup was based on the electrical stimulation systems protocol tested previously [50]. The device was built as displayed in Figure 1 using carbon rod electrodes, placing on the cover of the culture plates six-well plates (Thermo Scientific, Non-Treated, Paisley, UK) [51]. Carbon rod electrodes (2-mm diameter) were placed between the scaffolds, and electrical contact between carbon rod and scaffolds was ensured using a multimeter (Keysight, U1461A, Santa Rosa, CA, USA). Afterward, the electrodes were completely isolated with silicone rubber (Dow Corning 732 FDA 177.2600, Midland, MI, USA) in order to generate DC flow from the electrode to the composite scaffolds. For the electrically stimulated samples (PCL and PCL/TrGO), alligator connectors were used to tighten the contact between the wires and the electrodes. Triplicate samples for ES (under 30 V) and control conditions (non-stimulated samples) were used for the assays with bacterial cells and hBM-SCs.

The experimental setup (bioreactors and scaffolds) was disinfected and sterilized by submersion into 5 mL of ethanol 70% *v/v* for 15 min, followed by a wash with sterile phosphate buffered saline (PBS) (Sigma Aldrich) and exposure to UV light within a biosafety cabinet for 30 min.

*S. aureus* (ATCC 25922) grown for 48 h in dynamic conditions in tryptic soy broth (TBS) (Merck, Germany) solution was used for the antibacterial assays. After sterilization, 200 µL of bacterial suspension adjusted to 0.5 McFarland was seeded in each scaffold [26], and after incubation, an ES was applied. Based on the literature and considering the electric current values to obtain favorable bioelectric effects in both bacteria and human cells, the DC range used was obtained using the simulation software NI Multisim (Appendix A) with the real electrical resistance value of the scaffolds and voltage applied. The scaffolds were then removed and placed in centrifuge tubes (Sterile Falcon tube BD, Germany) and washed with 500 µL of fresh TBS three times. The final suspension was homogenized by a vortex agitator and ultrasonic bath for 3 min [52]. Aliquots of 100 µL, as well as with 10^−1^, 10^−2^, and 10^−3^ dilutions were collected and seeded onto tryptic soy agar (TSA) (Merck, Germany) plates (in triplicate for each sample and dilution) and incubated for 24 h at 37 °C to allow the growth of bacterial colonies or colony-forming units (CFU) [21,23,26]. For a complete evaluation of the bioelectric effect produced under ES, the same methodology was used to obtain the CFU in the solution once the scaffolds were removed from the bioreactor.

Cell viability under DC was tested using the Resazurin assay. Commercial hBM-SCs (Lonza, Lonza, Walkersville, MD, USA) were cultured in growth medium Dulbecco’s modified Eagle’s high-glucose (DMEM) (Sigma-Aldrich, Poole, UK), supplemented with 10% fetal bovine serum (FBS) (Sigma-Aldrich, UK) and 1% antibiotic antimycotic solution (Sigma-Aldrich, Poole, UK) at 37 °C and 5% CO_2_ in humidified atmosphere. A density of 1.5 × 10^5^ cells/scaffold were seeded in 4 mL of growth medium, which completely covered each scaffold, and were then incubated for 24 h. The growth medium was then changed 4 h before beginning the ES assays. A DC power supply (BK Precision, CA, USA) was used for ES assays under the same conditions as the antibacterial assays (30 V-3 h). After 18 h, scaffolds were removed from the bioreactor and 2 mL of 10% Resazurin sodium salt solution (0.125 mg/mL in PBS), (Sigma-Aldrich, Poole, UK) in growth medium was dropped on each scaffold using six-well plates as described by the authors of Reference [51]. After 2 h of incubation, the samples were read in a Synergy II plate reader (Biotek Instruments Ltd., Winooski, VT, USA) at 544-nm excitation and 590-nm emission.

### 2.5. Statistical Analysis

As an analysis of the variance, both one- and two-way ANOVA were used, and statistical differences were assessed by ANOVA with Tukey’s multiple comparison test. The Graph Pad-Prism 8.0.2. was used with a significance level of alpha 0.05 to evaluate the statistical significance with test values *p* < 0.05, *p* < 0.01, and *p* < 0.001.

## 3. Results and Discussions

### 3.1. Characterization of Graphene-Based Particles

GO and TrGO particles were characterized by different techniques in order to confirm the particle structure. X-ray diffractograms of the different particles are displayed in Figure 2a. The characteristic peak of crystalline graphite was obtained around θ ≈ 26° (002), indicating an interlaminar separation distance of 0.33 nm. After the oxidation process, the characteristic reflection from GO was obtained around θ ≈ 13°, indicating an increase in the distance between the graphene sheets to 0.73 nm due to the formation of functional groups during the oxidation process [42]. After thermal reduction, the diffraction spectra of TrGO particles showed a small and broad peak at around 25°, indicating a material without laminar graphite order corresponding to random platelets of corrugated structure [53]. Due to the gases released during the thermal reduction of GO functional groups, such as the hydroxyl and epoxy groups [41], pressure is generated between the stacked graphene layers which is higher than the necessary for its separation producing a volumetric expansion of up to 300 times, resulting in TrGO layers of low apparent density [54]. The presence of chemical functional groups in GO and TrGO particles was confirmed by narrow XPS scans as displayed in Figure 2b. Bands assigned to C sp^2^ from ordered carbon bonds of the graphene structures were found in both samples. However, the bands showed different relative intensities. In particular, the C1s band increased dramatically from ~47% (GO) to ~76% (TrGO), confirming the effective thermal reduction of GO particles [55]. Oxygenated groups contribute to structural defects in graphene sheets [43], as revealed by Raman spectroscopy analysis (see Figure 2c,d). These changes in the carbon structure of GO and TrGO particles were quantified by the ratio between the intensity of sp^3^ sites of disorder or structural defects (D band at ~1350 cm^−1^) and the intensity of sp^2^ sites (G band at ~1580 cm^−1^) in the carbon material surface from the Raman spectra (I_D_/I_G_) as displayed in Figure 2d. This ratio (I_D_/I_G_) is currently used as an indicator of defects in graphene structures arising from functional groups as compared with carbon array. In this study, intensity ratios of 0.97 and 0.87 were obtained for GO and TrGO, respectively, confirming that thermal reduction at 1000 °C decreased the amount oxygenated groups or structural defects [56].

### 3.2. Characterization of the Composite Materials

As the processes involved in the preparation of both the composite material and the scaffolds were carried out at temperatures above the melting temperature of the PCL polymer to compensate its high viscosity, it was important to obtain the thermal degradation profile of the samples using TGA. In particular, the initial degradation temperatures were determined at 5% weight loss (T_5%_). The initial thermal degradation temperatures of pure PCL were T _air 5%_ = 325 °C and T _nitrogen 5%_ = 362 °C under air and nitrogen atmospheres, respectively. The thermal degradation begins with the rupture of the polymer chains by the ester pyrolysis reaction releasing CO_2_, H_2_O and carboxylic acid. Chain scission or cleavage can occur randomly along the polymer [57]. Under both conditions, the TrGO filler improved the thermal degradation (T _air 5%_ = 335 °C and T _nitrogen 5%_ = 375 °C). Similar to other 2D fillers, it can be inferred that the degradation temperature was reduced due to the loss of chain mobility due to the action of TrGO sheets dispersed in the polymer by the barrier effect [58,59].

In addition, there was no loss of mass during the thermal treatment used for obtaining composites (melting mixture at 90 °C) or during the manufacture of scaffolds (220 °C) as shown in the TGA curve Figure 3a. Raman spectroscopy was carried out to PCL and PCL/TrGO films to analyze the interactions between the polymer matrix and the filler. Both samples presented the typical bands from PCL associated with C = C, CH_2_, and C = O bonds (Figure 3b).

In the polymer composite, the formation of chemical bonds between the TrGO and the PCL was not observed, and the overlapping and diminishing bands of the PCL could indicate that there was either electrostatic interactions or physical adsorption between the polymer and conductive particles [60]. Figure 4a shows representative photographs of the scaffolds as recorded by the 3D printer camera after finishing each layer. From these pictures, it is concluded that the printability and scaffolds characteristics were not affected by the presence of the nanofiller.

Figure 4b, shows optical pictures of the finished scaffolds, confirming the flexibility of the sample after processing even with the TrGO filler. SEM was further performed, allowing the measurement of the pore size of the 3D-printed scaffold and the quality of the impression. Figure 5a,b shows SEM images of the PCL scaffold from the top view, where a small triangular pore dimension of 300 ± 50 µm can be mainly observed. PCL/TrGO scaffolds presented pore sizes of 325 ± 150 µm (Figure 5c,d). The differences in the pore size can be explained by taking into account that the printing parameters were not the same for each sample. The filament with conductive particles was difficult to adhere to both the different printing surfaces tested and the previously printed layers of the material, so it was necessary to reduce the print velocity and increase the pressure to obtain the 3D-printed scaffolds.

After the melt mixing process, the composite films presented an electrical conductivity of (1.8 ± 4.5) × 10^−6^ s/m. At this concentration, the sample was above the percolation threshold. To check that electrical properties was retained after the 3D-printing process, the electrical resistance value of the resulting 3D-printed scaffolds was measured to calculate the volumetric electrical conductivity by considering the thickness of the five-layer scaffolds according to Reference [48]. The average conductivity value of the 3D-printed scaffolds with 10% wt of TrGO particles was (6.8 ± 0.36) × 10^−5^ s/m. The differences in conductivity values between films (2D) and 3D printing scaffolds were due to the melting process involved in obtaining the scaffolds. The temperature and print pressure produced TrGO sheet reorganization in the PCL matrix, so the final value of the conductivity varied slightly.

With the real values of electrical resistance and voltage applied, it was possible to obtain the ES conditions (Appendix A). Applying 30 V generated a DC current of ≈90 ± 11 µA in the scaffolds, which was delivered to the bacterial and human cells attached to the material surface and in contact with the culture media.

Surface chemistry is essential for bacterial and cell adhesion, as it is widely reported that by decreasing the hydrophobicity of the surface, the adhesion of *S. aureus* can also be decreased [61,62,63,64,65,66]. By incorporating 10% wt of TrGO particles into the PCL polymer matrix, the hydrophobicity of the material decreased significantly from an average value of 113.3 ± 10° to 74.8 ± 9° (Figure 6a). With the TrGO particles, hydrophilic functional groups were incorporated into the polymer composite, as previously discussed in XPS analysis (Figure 2a).

The roughness of biomaterial at the nanometric level was also analyzed due to the implications for human cell and bacterial adhesion [62,63]. In bone cells, nanoscale morphology has been used to control cell growth inducing bone tissue formation, bone recovery, and even osseointegration [61,64]. The roughness values were similar between both scaffolds as displayed in Figure 6b. AFM micrographs of each sample are shown in Figure 6c. However, a considerable difference in surface topography was observed, thus complementing the discussion of the results in both human cells and in bacteria. The Rsk and Rk values from the surface roughness distribution allowed a topographic characterization. The topographic evaluation was further carried out using values of Rsk > 0 and Rk > 3. On the other hand, a surface with deep valley and wide peaks was present at the values of Rsk < 0 and Rk < 3 [67], as shown in graphs from Figure 6c. With TrGO particle incorporation in PCL polymer matrix, the surface topography was modified, and sharp and narrow peaks were obtained as displayed in the topographic image of Figure 6c. In addition, the Rsk and Rsk values obtained were consistent with the topography studies performed on different types of surfaces reported in the literature [67,68,69].

### 3.3. Antibacterial Behavior under ES

Antibacterial tests were first carried out without ES in order to analyze the effect of the TrGO in the antibacterial behavior of the composite. The PCL/TrGO scaffolds displayed a bacterial reduction of 26% as compared with pure PCL, which was attributed to the hydrophobicity decrease in the printed composites, as displayed in Figure 7a. This was due to the functional groups of hydrophilic of TrGO particles (C = O and C-O) after the thermal reduction of GO [54]. Bacteria with hydrophobic characteristics, such as *S. aureus*, contain hydrophobic proteins in their cell walls, such as carboxylic phosphatase and teichoic acids [63] so their adhesion prefers the hydrophobic surfaces [70]. Another important factor in bacterial adhesion to biomaterials is the morphology of surface topography. Based on the topographic analysis of the scaffolds shown in Figure 6, the narrow and sharp peaks obtained on the PCL/TrGO scaffolds surfaces contributed to the detachment of bacteria without ES [71,72]. Spherical-shaped bacteria, such as coccus, readily adhered to the soft surfaces (R_sk_ < 0, pure PCL scaffolds) because they were less deformable and failed to adhere to surfaces with sharp peaks, such as Bacillus bacteria (*E. coli*) [71,72].

The assays with 3 h of ES demonstrated the bactericidal effect of the electroactive PCL/TrGO scaffolds under 30 V (DC current ≈ 90 ± 11 µA in the scaffold) during 3 h of ES using the experimental setup reported previously [73,74]. Similar conditions of ES were reported to eliminate *S. aureus* bacteria using stainless steel electrodes directly applied into the culture plates [27]. The carbon rods were completely isolated in this study by the biocompatible silicone, avoiding direct contact with the culture media and ROS production from electrolysis. Thus, ROS was not considered in the discussion of the bioelectric effect for both the assays with *S. aureus* and human cells. This was further confirmed through the pH indicators and chlorine test papers, which indicate that the electrochemical reactions between the electrodes and culture media were negligible (Appendix A). The presence of ROS in the level lower than our experiments detected could also contribute to the antibacterial behavior under ES.

In the pure PCL scaffold (electrically nonconductive) under ES, a reduction of 45% in the bacterial colonies attached to polymer scaffolds was obtained with respect to PCL at 0 V. This bacterial reduction was due to the electrophoretic effects associated with EF produced between the electrically isolated carbon electrodes. *S. aureus* bacteria have a single-membrane that is sensitive to external EF [28], which is able to produce structural damage, such as a loss of integrity in the membrane or irreversible electroporation. Thus, this provokes an intracellular content leakage that decreases bacterial viability [21,23]. In the electroactive PCL/TrGO scaffolds under ES, 100% of bacterial colonies were eradicated under ES, as displayed in Figure 7a.

One of the antibacterial mechanisms is associated with the electron flow through the electroactive conductive biomaterial, producing an antiadhesive effect by the electrophoretic forces, which increase bacterial mobility generating desorption on 3D-printed conductive scaffolds [75]. *S. aureus* can easily donate electrons to an external conductive surface producing electron transfer from the bacterial membrane under DC [28]. Therefore, the bacteria are pushed away from the biomaterial surface and detached by electrons transfer from the bacteria membrane [27,32], leading to membrane autolysis and cell disruption [21]. This bactericidal effect was confirmed by analyzing CFU bacteria from the bioreactor solution, as shown in Figure 7b. In the case of the stimulated samples, there was a significant decrease in bacteria in the culture medium solution, which indicates that it produced a bactericidal effect associated with the effect of ES. For the PCL scaffolds culture medium, there was a 49% bacterial decrease when compared with the control (non-ES) due to the effect that EF produced between the electrodes as previously mentioned. In the culture medium of the porous conductive scaffolds, the electrical current, which passed through the PCL/TrGO scaffolds for 3 h, increased the bacterial reduction to 89%. The hypothetical antibacterial mechanisms are illustrated in Figure 7c.

### 3.4. Cell Viability of hBM-SCs under Antimicrobial Stimulation Regime

The same ES conditions with bacterial assays (30 V-3 h) were used to evaluate the cellular viability of hBM-SCs adhered to the electroactive scaffolds. The metabolic activity assays using resazurin sodium salt were tested, as shown in Figure 8. The hydrophobicity of PCL is considered a disadvantage for tissue engineering because it hinders cell adhesion, and surface treatments are currently necessary, such as treating with sodium hydroxide (NaOH) solutions to reduce their hydrophobicity and increase cell adhesion on the polymer [76,77]. Therefore, the incorporation of 10% wt of TrGO particles, which decreased the hydrophobicity of the PCL composite, increased the cytocompatibility of the scaffold due to the physical-chemical interactions between the cells and surface of the scaffold based on the observation that the adhesion of hBM-SCs increased by two-fold.

Cell adhesion on the electroactive composite after applying ES of 30 V (producing a DC current of ≈90 ± 11 µA in composite scaffolds) was even higher when compared with pure PCL, confirming the positive effect of ES on the electroactive polymer. For decades, it has been hypothesized that electrical signals are transduced through the calmodulin protein and calcium present inside the cells, which cause cellular changes under external electrical stimuli, either via direct or capacitive ES [5]. This mainly affects the cell membrane by increasing the concentration of calcium ions ($Ca^{2+}$) passing through [78] and the level of prostaglandin E2 by activating the ion channels on the cell membrane [5,73]. Specifically, ES depolarizes the cell membrane, which activates and opens ion channels via voltage and/or the alteration of ion fluxes, such as $Ca^{2+}$, thereby contributing to the stimulation of cellular locomotion, or electrophoretic, or electroosmotic effects and leading to the redistribution of cell membrane components [78]. These mechanisms support the evidence that shows the great influence of ES on the growth and development of nerve and bone cells, as well as the process of wound healing and angiogenesis [16,78]. The results from this study confirm that the ES regime required to obtain an antibacterial behavior not only does not generate any cytotoxic effect on human cells, but also produces greater cell viability and when electroactive scaffolds are used.

## 4. Conclusions

Conductive polymeric composite scaffolds based on PCL with TrGO filler were successfully 3D printed, allowing a selective effect of ES on both bacterial growth and human cell viability to be studied. The presence of TrGO in the PCL scaffold partially decreased bacterial growth when compared with pure PCL without ES. Remarkably, the application of electrical stimuli (30 V-3 h) on the 3D-printed electroactive scaffolds completely eradicated bacterial growth on the scaffold surface, while pure PCL scaffold still possessed bacterial attachment after ES. The incorporation of TrGO into the PCL scaffolds increased the viability of hBM-SCs, which was associated with the reduction in the hydrophobicity of the scaffold. Notably, under the same ES regime used during the bacterial tests, higher cell viability was observed in the electroactive scaffolds when compared with pure PCL. These results have widened the design of novel electroactive biomaterials for bone tissue engineering which are not only able to eliminate bacteria, but also promote mesenchymal stem cell growth and viability under ES.

## Figures and Tables

**Figure 1 nanomaterials-10-00428-f001:**
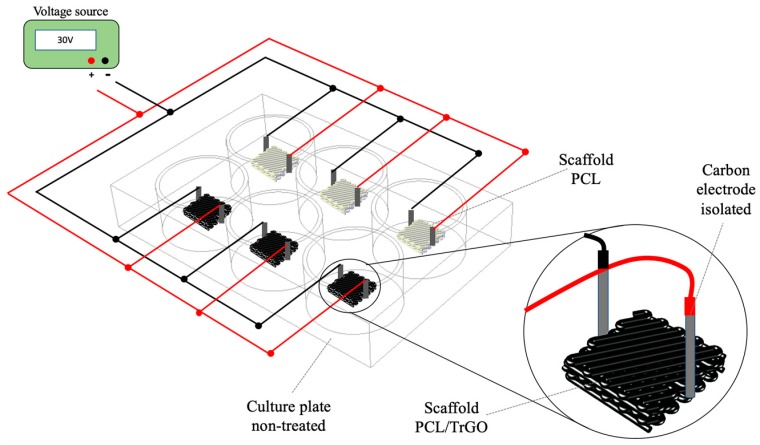
Experimental setup used for the assays with bacterial cells and hBM-SCs under 30 V-3 h.

**Figure 2 nanomaterials-10-00428-f002:**
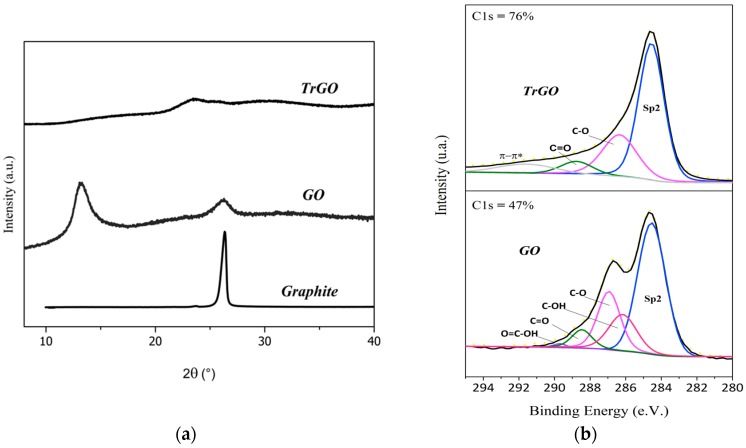
Characterization of graphene-based particles: (**a**) X-ray diffractogram; (**b**) pseudo-Voigt adjustment from high-resolution XPS spectra of TrGO and GO particles; (**c**) Raman spectra; and (**d**) intensity ratio (I_D_/I_G_) by Lorentzian curve adjustments.

**Figure 3 nanomaterials-10-00428-f003:**
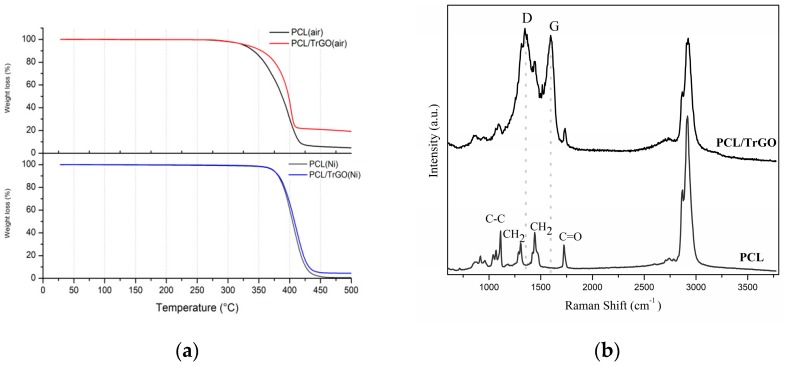
(**a**) Thermogravimetric analysis (TGA) curves in air and nitrogen atmosphere; (**b**) Raman spectra of PCL/TrGO composite and PCL materials.

**Figure 4 nanomaterials-10-00428-f004:**
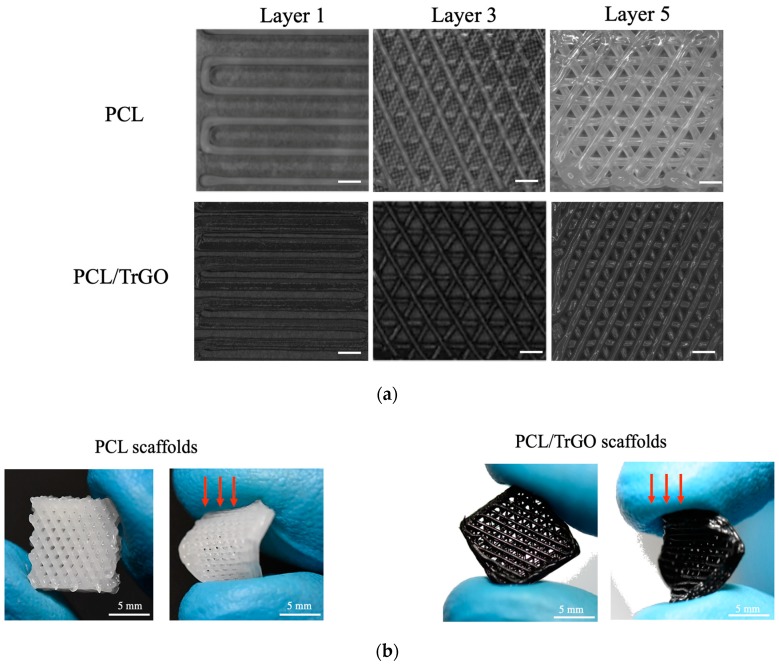
(**a**) Images captured from the scaffold printing process of layers 1, 3, and 5, scale bar 300 µm; (**b**) optical photographic of the scaffolds frontal view (left) and applying qualitative stress (right).

**Figure 5 nanomaterials-10-00428-f005:**
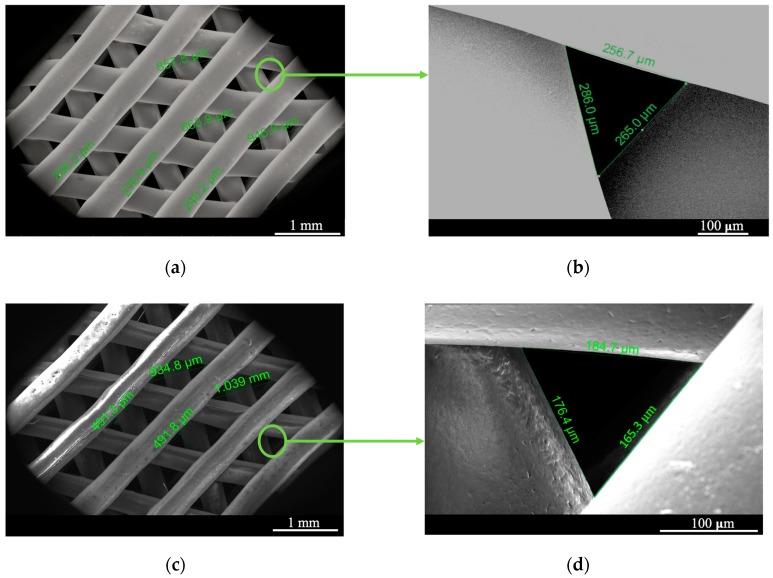
(**a**) Top view of PCL scaffold; (**b**) details and size of a pore; (**c**) top view of the scaffold with the conductive TrGO particles; (**d**) detailed image of a scaffold pore is shown.

**Figure 6 nanomaterials-10-00428-f006:**
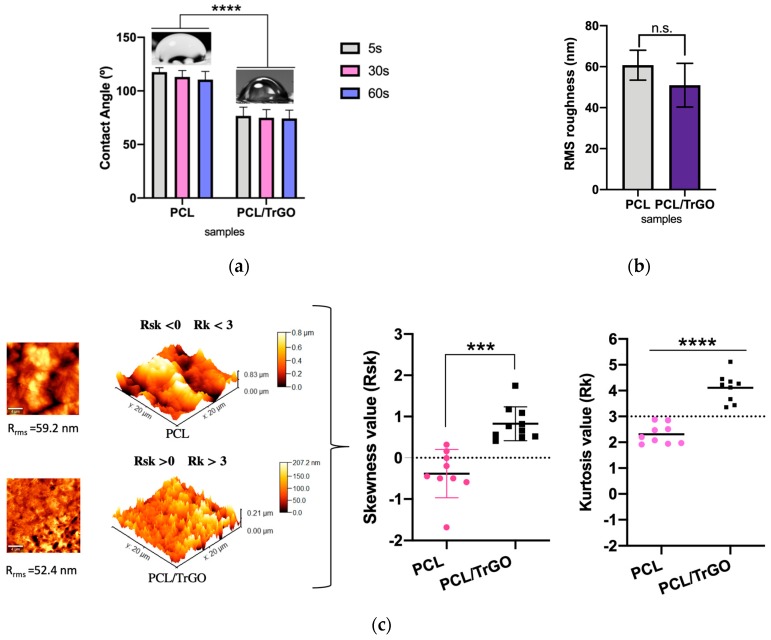
(**a**) Contact angle measurement for PCL and PCL/TrGO scaffolds. **** denotes a significant difference with a value *p* < 0.0001 between the samples indicated (n = 5 independent tests performed for each time spent); (**b**) RMS roughness values of PCL an PCL/TrGO scaffolds, n.s. denotes nonsignificant difference between the samples (n = 5 different zones of measurements); (**c**) AFM micrograph of surface roughness and topography of PCL scaffolds and PCL/TrGO scaffolds (n = 5 independent measurements) (left), graphs of skewness (Rsk) and kurtosis (Rk) values of PCL and PCL/TrGO samples (right). *** and **** denote a significant difference with a value *p* < 0.001 and *p* < 0.0001, respectively, between the samples indicated.

**Figure 7 nanomaterials-10-00428-f007:**
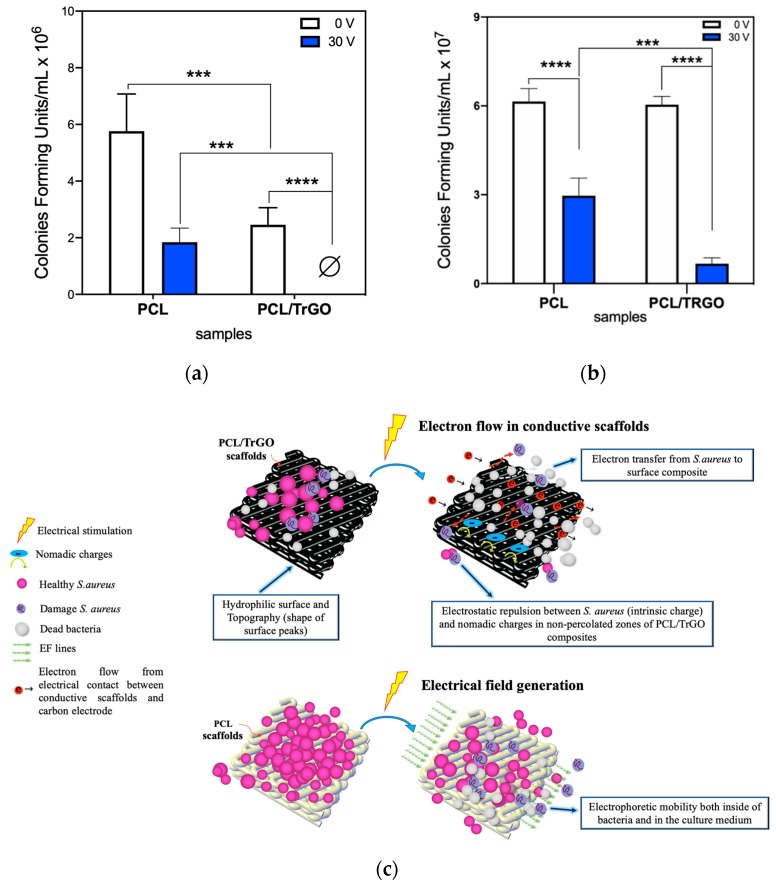
(**a**) Colony-forming units (CFU) graph of the antibacterial tests with (30 V) and without electrical stimuli (0 V) for 3 h. The symbol Ø on the PCL/TrGO at 30 V denotes nondetectable values, (n = 4 independent experiments in triplicate); (**b**) bacterial CFU in the culture media (scaffolds removed from bioreactor) after 3 h of incubation at 0 V and 30 V. ** and *** denote a significant difference with a value *p* < 0.001 and *p* < 0.0001, respectively, with the control sample PCL 0 V and between tche samples indicated, n = 3 independent experiments were performed in triplicate per sample; (**c**) schematic representation of the possible mechanisms of antibacterial effects at 0 V for PCL/TrGO and under conditions of electrical stimulation (ES) at 30 V for scaffolds of PCL and PCL/TrGO.

**Figure 8 nanomaterials-10-00428-f008:**
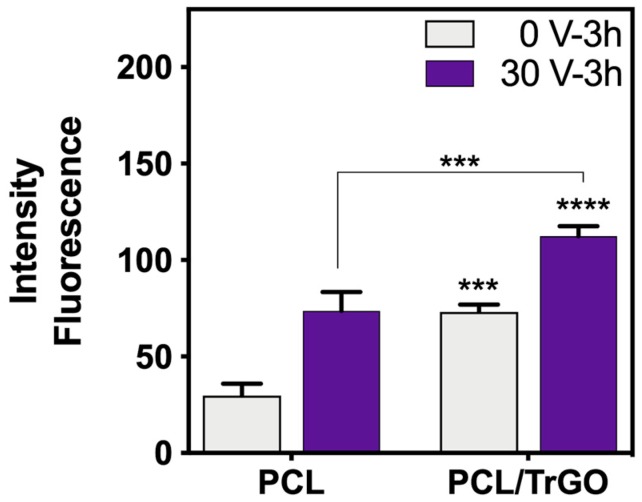
Metabolic activity graph of hBM-SCs obtained from the resazurin assay without ES (0 V) in gray and ES (30 V) in purple with 3 h of ES. *** and **** denote a significant difference with a value *p* < 0.001 and *p* < 0.0001, respectively, between the samples indicated, and a value *p* < 0.001 denotes the control sample PCL 0 V. n = 3 independent experiments were performed in triplicate per plate at three days from seeding cells in/on the scaffolds.

**Table 1 nanomaterials-10-00428-t001:** Printing parameters for obtaining Polycaprolactone (PCL) and PCL/Thermally Reduced Graphene Oxide (TrGO) scaffolds.

MATERIAL	PCL	TRGO
Temperature	170 °C	220 °C
Atmosphere	Nitrogen	Nitrogen
Pressure	4.5 bar	6 bar
Robot Head Velocity	0.5 mm/s	0.3 mm/s
Pre-flow	2 s	2.5 s
Post-flow	1.5 s	2 s
Needle inner Diameter	0.9 mm-20 g	0.9 mm-20 g
Warm up Time	50 min	60 min

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
