# Peer review of "Electroactive 3D Printed Scaffolds Based on Percolated Composites of Polycaprolactone with Thermally Reduced Graphene Oxide for Antibacterial and Tissue Engineering Applications"

_nanomaterials, 2020, doi:10.3390/nano10030428_

Round 1

Reviewer 1 Report

The manuscript entitled “Electroactive 3D Printed Scaffolds based on Percolated Composites of Polycaprolactone with Thermally Reduced Graphene Oxide for Antibacterial and Tissue Engineering Applications” described the fabrication, characterization, antibacterial effects and cytocompatibility of a 3D printed PCL/TrGO scaffold, with multiple techniques applied including 3D printing, XRD, XPS, TGA and cell assays, etc. It’s well-structured and the topic is novel and interesting combining 3D printing and the antibacterial effects of electric stimulation on graphene-doped biopolymers. The authors also describe several potential mechanisms to explain for the antibacterial effects. I recommend the paper for publication with minor comments; 

Scale bar is needed for Fig. 4. Line 316-317; The authors said the bacterial reduction is 26% from Fig. 7a. How did the authors calculate this from the CFU counting? The authors claim that the bacterial adhesion and the hydrophilicity/hydrophobicity of the surface as the driving forces for the antibacterial effect. Yet, there are several reports on the generation of ROS from surface charge and ES to induce antibacterial effect. The authors should consider this ROS effect also. Result in Fig. 7b is a simple calculation from Fig. 7a and thus, they present the same data to some degree. I suggest the authors change Fig. 7b to SEM image revealing the death of bacteria on the scaffold after US, which will better support the mechanism they came up with. In the current manuscript, there is no direct evidence to support the mechanisms. Finally, English should be checked carefully and thoroughly as there are still some minor typos there.

Reviewer 2 Report

The manuscript of Angulo-Pineda et al describes the potential use of scaffolds composed of PCL with TrGo in cell culturing, providing strong antibacterial effects. The paper is well structured and data clearly presented and discussed. The expertise of the research group is obvious and consistent with their previous publications. A few comments for improvement:

How were 30 V selected as optimal ES condition? Did the authors perform kinetics? use less or higher (less probable) Vs? please explain. Lines 260-262: I find the pore size differences between PCL and PCL/TrGO scaffolds quite odd. Particularly in PCL/TrGO, the SD of 150 um, meaning you have pores of ~200-450 um. This is explained in text by different printing parameters used. My questions are: (a) how did the same cells (I refer to hBM-SCs) adhere and proliferate in a scaffold with so different pore sizes? I assume that some areas with wider pores would be “empty” of cells, meaning that if let incubate for longer periods maybe the cells would die in overpopulated areas; and (b) what means different printing parameters? Why these cannot be the same? What was the rational for selecting hBM-SCs? Did the authors test other cell types using the same scaffolds? eg. fibroblasts, adherent cancer cells? I would appreciate a photograph or better more (preferably SEM) of how bacteria and hBM-SCs “look” upon adhesion on the scaffolds. These can be added as suppl. material. Figure S5: statistical significance should be tested between 0 and 30 V per group, i.e., on PLC (shown ***) and PCL/TRGO (not shown), please add. I would also check statistics in the 30 V groups PLC vs PCL/TrGO. Details missing in figures: fig 5d, value in vertical right pore area; fig. 6c reverse axes in kyrtosis value (RK) diagram and add statistics in both graphs; fig. 7c, electric field lines in PCL/TrGO scaffolds?; fig. 8, measurement units of intensity fluorescence, is it MFI? Check abbreviations, full wording should be written the first time in text followed by abbreviation in parenthesis and only abbreviation thereafter. Eg. line 149 DC, 177 CFU. Check for grammar, eg. lines 46, 69-70, 107 (rpm), 169-171 (verb missing), 299, 325, 355 (bacterial), 358, 359 (are, of figure) 371, 402.

Author Response

This manuscript is a resubmission of an earlier submission. The following is a list of the peer review reports and author responses from that submission.